# Learning Global Representation from Queries for Vectorized HD Map Construction

Shoumeng Qiu [* 1 2]   Xinrun Li [* 3]   Yang Long [3]   Xiangyang Xue [1]   Varun Ojha [4]   Jian Pu [1]

## Abstract

The online construction of vectorized high-definition (HD) maps is a cornerstone of modern autonomous driving systems. State-of-the-art approaches, particularly those based on the DETR framework, formulate this as an instance detection problem. However, their reliance on independent, learnable object queries results in a predominantly local query perspective, neglecting the inherent global representation within HD maps. In this work, we propose **MapGR** (**G**lobal **R**epresentation learning for HD **Map** construction), an architecture designed to learn and utilize global representations from queries. Our method introduces two synergistic modules: a Global Representation Learning (GRL) module, which encourages the distribution of all queries to better align with the global map through a carefully designed holistic segmentation task, and a Global Representation Guidance (GRG) module, which endows each individual query with explicit, global-level contextual information to facilitate its optimization. Evaluations on the nuScenes and Argoverse2 datasets validate the efficacy of our approach, demonstrating substantial improvements in mean Average Precision (mAP) compared to leading baselines. Code is available at github.com/skyshoumeng/MapGR.

## 1. Introduction

Online high-definition (HD) map construction is a crucial task in autonomous driving (Liu et al., 2023; Yuan et al., 2024; Chen et al., 2024; Hao et al., 2024), as it aims to provide high-precision perception of map elements essential for safe and efficient navigation. Unlike traditional offline HD maps (Shan & Englot, 2018; Shin et al., 2025), which require extensive pre-collection and manual updates, online HD map construction allows autonomous vehicles to construct local maps on the fly by leveraging sensors such as LiDAR and cameras, resulting in more easily adapting to changing road conditions such as construction zones, lane modifications, and unexpected obstacles. Many prior online HD map construction methods have treated this task as a semantic segmentation problem in Bird's-Eye-View (BEV) space (Gosala et al.; Li et al., 2022b; Philion & Fidler, 2020; Zhou & Krähenbühl, 2022). However, their effectiveness is limited by the extensive post-processing needed to extract vectorized map representations. To overcome these limitations, recent research has shifted toward DETR-like frameworks (Ding et al., 2023; Qiao et al., 2023; Yu et al., 2023; Liu et al., 2024a), leveraging learnable queries for vectorized instance detection.

Unlike conventional object detection scenarios where targets exhibit independent spatial distributions (characterized as 'spiky' distributions), High-Definition (HD) maps manifest distinctive spatial continuity in the BEV space, which we characterize as 'streak' distributions. This fundamental distributional divergence presents significant challenges in the direct application of the DETR (Detection Transformer) framework, as its architecture was primarily optimized for object detection tasks that inherently accommodate spiky distributional patterns. To address this issue, existing approaches introduce manual instance partitioning for maps, where each instance is represented by a set of discrete points and treated as an independent object, as in the object detection task. This transformation enables the direct application of DETR-like framework to HD map reconstruction tasks, with training objectives designed to encourage the model outputs to approximate the partitioned instances. However, this manual partitioning approach has several obvious limitations. First, information loss is inevitable during the sampling process. For example, representing each instance with a finite set of points may result in the loss of local road structure details (Zhang et al., 2023a). Additionally, independent optimization of each instance overlooks the spatial relationships between instances as well as their structural dependencies in the global map, potentially leading to suboptimal learning in optimization.

---
[*]Equal contribution [1]Fudan University [2]Bosch, China [3]Durham University [4]Newcastle University. Correspondence to: Yang Long <yang.long@durham.ac.uk>, Jian Pu <jianpu@fudan.edu.cn>.

*Proceedings of the 43rd International Conference on Machine Learning*, Seoul, South Korea. PMLR 306, 2026. Copyright 2026 by the author(s).

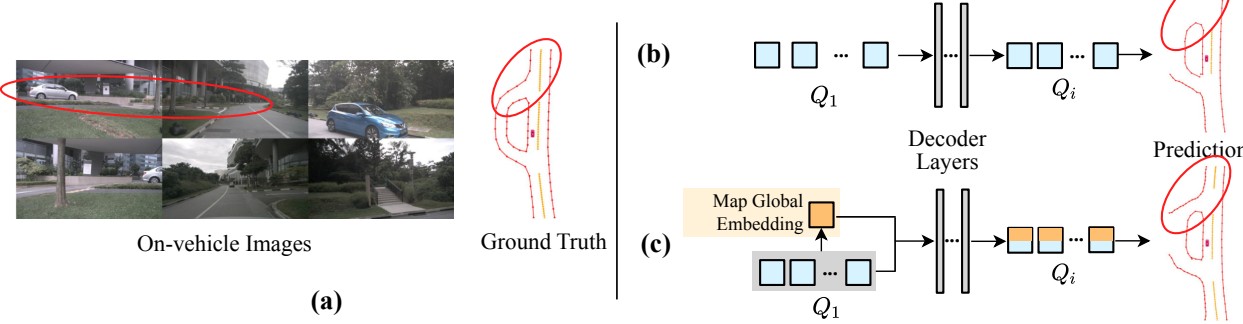

*Figure 1.* (a) Multi-view images from on-board sensors. (b) A conventional DETR-like HD map construction pipeline. (c) Our proposed global representation learning of queries for the map construction task significantly improves query distribution from the initial to the final decoder layers. This improvement leads to smoother and more consistent curvature changes in instances, ensuring better alignment with the global structure. $Q_i$ represents the query set from the $i$-th decoding layer. The red box marks a region where query distribution improves significantly.

To address the above limitations, we introduce a method that enables the model to learn a global HD map representation directly from all object queries, which is subsequently leveraged to facilitate the learning of each individual query. Specifically, our approach consists of two key components: a Global Representation Learning (GRL) module and a Global Representation Guidance (GRG) module. The GRL module aims to learn a global map representation from all queries, then the representation is used to predict a holistic, rasterized representation of the map, which is supervised by the Ground Truth (GT) map. The GRG module aims to enhance the information of each local query by incorporating the global representation learned from the GRL module. The global representation encompasses not only the information of all queries but also the information derived by the GRL module based on all the queries as inputs. By integrating the global representation into queries, each query can be optimized individually while also maintaining a global perspective at the same time. The main differences between our method and previous approaches, as well as our advantages in map reconstruction results, are presented in Figure 1. Extensive experiments on public challenging HD map construction datasets, including both nuScenes (Caesar et al., 2020) and Argoverse 2 (Wilson et al., 2023) demonstrate that the proposed method achieves better mean average precision (mAP) while maintaining good efficiency. The main contributions of our work can be summarized as follows:

- We propose an effective and efficient approach for HD map construction based on global representation learning across all queries, formulated as a plug-and-play module seamlessly compatible with mainstream methods, including the MapTR series.

- We design a Global Representation Learning module to enhance the global distribution learning of queries by aggregating local queries into a global embedding,

which is then supervised by the rasterized GT map distribution.

- Additionally, we further propose an effective Global Representation Guidance module, which can guide the optimization of individual queries through the global information of maps.

- Extensive experiments on nuScenes and Argoverse2 confirm that our approach significantly improves performance across diverse baselines, achieving state-of-the-art results.

## 2. Related works

### 2.1. Query-based Object Detection

DETR (Misra et al., 2021) introduced a class of end-to-end query-based models that treat object detection as a set prediction problem. During the training of DETR, a predefined number of object queries are matched to either ground truth or background by solving the Hungarian Matching problem. Multiple decoder stages iteratively refine the queries, similar to Cascade RCNN, with each intermediate stage being supervised by the matching results. In recent years, many algorithms have been developed based on the concept of DETR. Deformable DETR (Zhu et al.) introduces a deformable attention module that addresses previous limitations and significantly enhances convergence speed by a factor of 10. Conditional DETR (Meng et al., 2021) separates object queries into content and spatial queries within the decoder's cross-attention module, enabling the model to learn a conditional spatial query from the decoder embedding. This facilitates the rapid learning of distinctive object boundaries in ground-truth data. Anchor-DETR (Wang et al., 2022) structures object queries as anchor points, allowing each query to focus on a specific region near its assigned anchor. Efficient DETR (Yao et al., 2021) enhances DETR's

performance by integrating a dense prior into the query mechanism. DAB-DETR (Liu et al., 2022) explores the role of object queries more deeply by directly utilizing anchor box coordinates as spatial queries to accelerate training. The model leverages spatial priors by adjusting the positional attention map based on the width and height of the bounding box. DN-DETR (Li et al., 2022a) further enhances the convergence speed and query-matching stability of DAB-DETR through a GT denoising mechanism.

## 2.2. Query-based Map Construction

VectorMapNet (Liu et al., 2023) is an end-to-end mapping approach that directly predicts vectorized maps from sensor data, avoiding rasterization and post-processing. It represents map elements as ordered polylines and treats their construction as a detection task. It uses ordered polylines and DETR models to detect 3D structures, outperforming centerpoint-based methods in HD map learning. MapTR (Liao et al., 2022) proposed a structured end-to-end framework for efficient online HD map construction. It introduces a permutation-equivalent modeling approach that represents map elements as point sets with equivalent permutations, improving shape accuracy and learning stability. By utilizing hierarchical query embedding, bipartite matching, and loss functions to supervise geometric structures. BeMapNet (Qiao et al., 2023) is an efficient HD-map modeling method using piecewise Bézier curves. It integrates geometric priors, models dynamic curves, and applies multi-level supervision through PCR-Loss. MapTRv2 (Liao et al., 2024) is an improved version of MapTR, featuring a structured end-to-end framework with hierarchical query embeddings and decoupled self-attention to enhance computational efficiency. Additionally, an optimized training strategy significantly boosts performance and convergence. Pivotnet (Ding et al., 2023) incorporates both subordinate and geometrical point-line priors into the network. MapVR (Zhang et al., 2023a) uses differentiable rasterization to improve vectorization accuracy and scalability. It also employs a rasterization-based evaluation metric to better detect small deviations and assess map vectorization performance. GeMap (Zhang et al., 2023b) explicitly models local geometric structures as shape attention and relation attention in the learning process. HiMap (Zhou et al., 2024) provides an efficient framework with a hybrid representation. A point-element interaction module helps predict precise point coordinates and element shapes by fusing information from both levels. MapQR (Liu et al., 2024b) proposes a novel online end-to-end map construction method based on scatter-and-gather queries. Combining with compatible positional embeddings facilitates point-set-based instance detection within DETR architectures. Although most DETR-like frameworks have achieved promising results, map reconstruction differs from object detection in

that it entails richer global information. Yet, this crucial aspect is often overlooked by mainstream approaches.

## 3. Methods

### 3.1. Overall Architecture of MapGR

Our method aims to construct HD map of the surrounding environment through multi-view images captured by cameras on the vehicles. The map is represented as multiple distinct instances, with each instance corresponding to a portion of the whole map. For example, an instance may represent a part of a lane. The combination of all instances forms the complete map. Each instance is represented as a set of points: $P = \{(x_i, y_i)\}_{i=1}^{l}$, where $l$ denotes the number of points used for each lane line and $(x_i, y_i)$ denotes the 2D coordinates. Additionally, the instance also contains class information, such as lane divider, pedestrian crossing, and road boundary. The overall workflow of the proposed method is shown in Figure 2.

Given the input surrounding images $Imgs = \{img_0, img_1, \ldots, img_{k-1}\}$, where $k$ denotes the number of images, typically $k = 6$ for the nuScenes dataset and $k = 7$ for the Argoverse 2 dataset. We first use a 2D image feature extraction network to get the image features and then project the 2D features into the 3D BEV space using Camera-to-BEV transformation, obtaining the BEV features $F_{bev} \in \mathbb{R}^{C \times H \times W}$, where $C$, $H$, and $W$ represent the channel dimensions, height, and width of the BEV features, respectively. We then decode the instance predictions from BEV features with a multi-layer transformer. The proposed GRL and GRG modules are integrated into the Transformer layer decoding process, facilitating more effective query learning. It can be seen (Figure 2) that our proposed method has minimal impact on the overall framework, making it a convenient plugin that can be easily applied to most of the current mainstream frameworks.

### 3.2. Global Representation Learning Module

#### 3.2.1. FORMULATION OF QUERY REPRESENTATION LEARNING

Given the multi-view images as inputs for the model, the map construction task seeks to minimize the discrepancy between the predictions $\mathcal{D}_{pred}$ and the ground truth $\mathcal{D}_{gt}$:

$$\min Dist(\mathcal{D}_{gt}, \mathcal{D}_{pred}), \tag{1}$$

where $Dist(,)$ is a distance function used to measure the distance between two inputs. $\mathcal{D}_{gt} = \{I_{gt}^0, I_{gt}^1, \ldots, I_{gt}^{m-1}\}$, where $I_{gt}^i$ is the partitioned instances from the overall GT map. $\mathcal{D}_{pred} = \{I_{pred}^0, I_{pred}^1, \ldots, I_{pred}^{n-1}\}$, where $I_{pred}^i$ denotes the prediction results of instance query $q_i$. Typically, $n > m$ to accommodate the varying number of GT samples

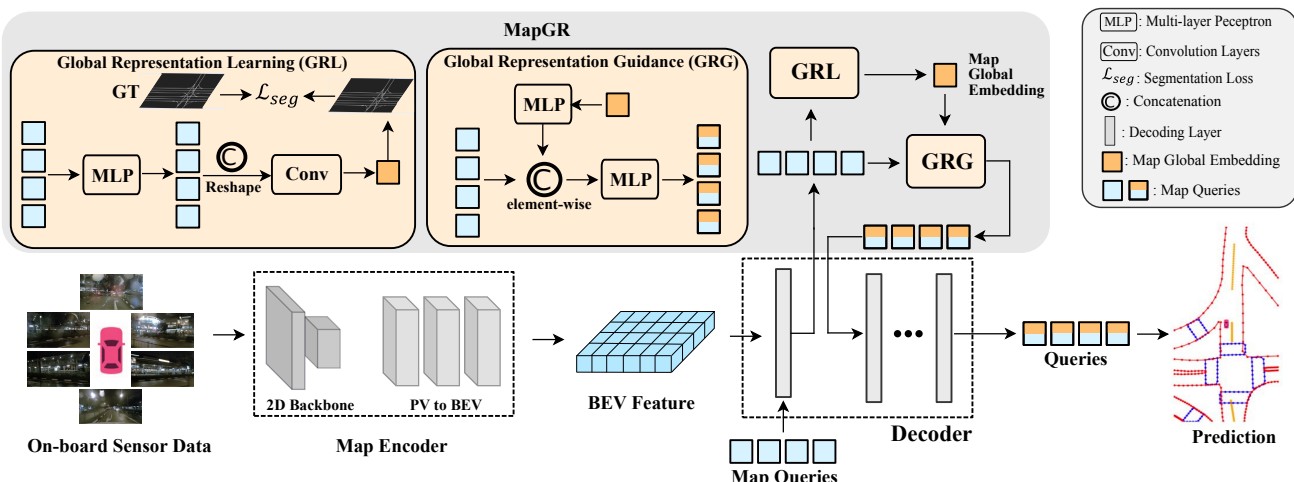

*Figure 2.* The details of our proposed method. The map encoder transforms multi-view images into a BEV embedding, while the decoder enables map queries to interact and extract information from the BEV embedding to decode vectorized map instances. The GRL module aggregates these queries into a global representation for the overall map distribution. The global representation is then used by the GRG module to enhance the query in the subsequent decoding process.

across different scenarios. The calculation of the distance between $\bar{\mathcal{D}}_{gt}$ and $\bar{\mathcal{D}}_{pred}$ involves matching ground truth samples to the predictions generated by the queries. The objective can be written as:

$$\min \sum_{I_{gt}^i \in \mathcal{D}gt} dist(I_{gt}^i, I_{pred}^{\pi(i)}), \qquad (2)$$

where $I_{gt}^i$ and $I_{pred}^{\pi(i)}$ represent the matched pair of a ground truth instance and its corresponding query prediction. $\pi(i)$ represents the index of query in the model's query set $Q = \{q_0, q_1, ..., q_{n-1}\}$ that is successfully matched to $I_{gt}^i$.

In this paper, we argue that learning from the individual instance presents certain limitations. First, representing each instance as discrete points may result in the loss of fine-grained details. Second, the instance generation process may introduce subjective factors, such as the map partition strategy, that may confuse the model learning. Lastly, instance-level constraints operate at a local scale, overlooking the global information of the map, potentially leading the query learning to fall into a local optimum.

To address the issues identified in the above analysis, we introduce an auxiliary task focused on learning a global representation from queries for the map. Rather than concentrating on individual instance learning, this task involves a function $\mathcal{F}(\cdot)$, that jointly processes all the queries $Q$ and then decodes into a global holistic map prediction $M_{pred}$, rather than predicting local map segments. The process can be expressed as: $M_{pred} = Decode(\mathcal{F}(Q))$, the predicted map $M_{pred}$ is directly supervised by the global GT map $M_{gt}$, eliminating the need for instance matching. The learning objective is to minimize the discrepancy between these two map representations:

$$\min D_{global}(M_{gt}, Decode(\mathcal{F}(Q))), \qquad (3)$$

where $D_{global}(,)$ is a distance measurement function that evaluates the similarity of the overall map.

### 3.2.2. MODULE DESIGN DETAILS

**Global Supervision from GT**: To realize the global representation learning in Equation 3, we first need to obtain the representation for the GT map $M_{gt}$. This is achieved by rasterizing the ground truth map and representing it in the BEV space. Each pixel in the BEV pseudo-image is denoted as $pixel_{\{i,j\}} \in \{0, 1\}$, where $i \in [0, H-1]$ and $j \in [0, W-1]$. Specifically, $pixel_{\{i,j\}} = 0$ indicates the absence of map element at that location, while $pixel_{\{i,j\}} = 1$ signifies the presence of a map element. We take this binary mask as $M_{gt}$ in this paper. In practice, we take into account not only the presence of map elements but also their semantic categories, such as pedestrian crossings and road boundaries. So the mask is multi-channel, denoted as $M_{gt} \in \mathbb{R}^{C \times H \times W}$, where $C$ represents the number of classes, with each channel corresponding to a distinct semantic category.

**Global Representation Construction from Queries**: In a standard DETR-like framework, instance predictions are generated from a set of queries $Q$, which will be refined through several transformer decoder layers. Ultimately, each query $q_i^k$ is independently passed through a Multi-Layer Perceptron (MLP) to produce a prediction for the individual instance. Therefore, to construct the global representation of the map, it is essential to consider all the queries. In the following, we provide a detailed description of the global representation construction process.

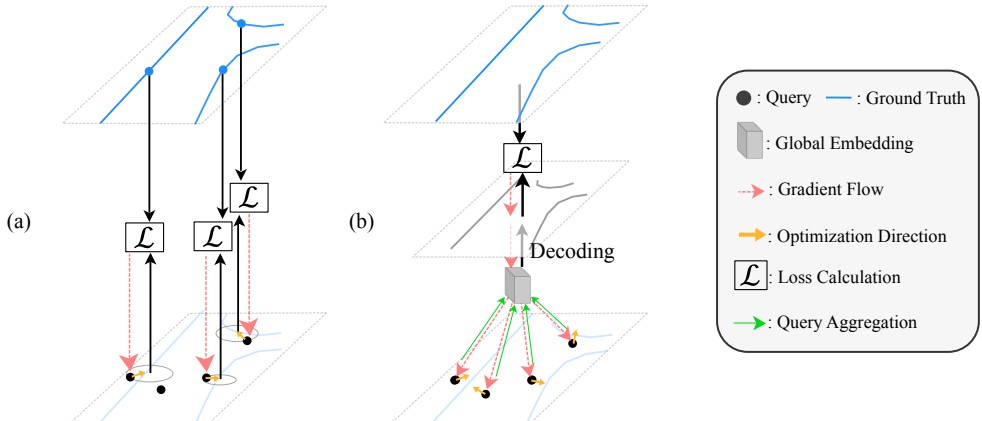

*Figure 3.* (a) Sampling and matching-based query learning. (b) Global representation aided query learning. It is evident that not all queries can be matched and obtain gradients. However, by leveraging global embedding to aggregate queries, all queries can obtain gradients derived from the global distribution prediction.

First, each query feature $q_i^k \in \mathbb{R}^{1 \times C}$ is individually projected and reshaped into a small 2D spatial feature map $\bar{\mathbf{q}}_i^k \in \mathbb{R}^{1 \times h \times w}$ using an MLP.

$$\bar{\mathbf{q}}_i^k = \text{Reshape}(\text{MLP}(q_i^k)) \qquad (4)$$

Then, these individual spatial feature maps are concatenated along a new dimension, stacking them to form a unified tensor $\bar{\mathbf{Q}} \in \mathbb{R}^{n \times h \times w}$ that aggregates information from all $n$ queries.

$$\bar{\mathbf{Q}} = \text{Concat}(\bar{\mathbf{q}}_i^1, \bar{\mathbf{q}}_i^2, \ldots, \bar{\mathbf{q}}_i^n) \qquad (5)$$

This aggregated tensor $\bar{\mathbf{Q}}$ is then processed by a lightweight convolutional neural network $\phi(\cdot)$ and upsampled via bilinear interpolation to match the target BEV resolution $(H, W)$. This yields the final predicted global map $M_{pred} \in \mathbb{R}^{C \times H \times W}$.

$$M_{pred} = \text{Upsample}(\phi(\bar{\mathbf{Q}})) \qquad (6)$$

With both $M_{gt}$ and $M_{pred}$ obtained, we choose to apply a standard Binary Cross-Entropy (BCE) loss between the predicted map $M_{pred}$ and the ground truth map $M_{gt}$ to enforce consistency between them. Thus the Global Representation Learning Loss, $\mathcal{L}_{global}$, is formulated as:

$$\mathcal{L}_{global} = \text{BCE}(M_{pred}, M_{gt}) \qquad (7)$$

The differences between our proposed global representation learning and the previous DETR-like approach are illustrated in Figure 3. As depicted, the previous method operates only on successfully matched predictions, with each prediction focusing exclusively on its paired ground truth (GT) instance during optimization. In contrast, our method propagates gradients to all instance queries in optimization, as the $\mathcal{L}_{global}$ is computed based on all queries.

It should be noted that in MapTR and subsequent methods based on MapTR, each point within an instance has a distinct point-level query for prediction, i.e., $q_i = \{p_0, p_1, \ldots, p_{l-1}\}$. In this scenario, to simplify the above process and reduce computational costs, we compute the mean of all point query features that belong to the same instance to form one feature representation $q_i = \text{Mean}(p_0, p_1, \ldots, p_{l-1})$.

### 3.3. Global Representation Guidance Module

The estimated $M_{pred}$ captures the global map information of the current scene. We argue that incorporating this global information into individual queries can enhance query learning for the following reasons: First, incorporating global information allows query learning beyond local perspectives and steers the optimization process from a more global viewpoint. Furthermore, although global information is derived from all individual queries, the global representation estimation model may be able to infer missing details or reconstruct incomplete regions of the map by leveraging existing predictions. Consequently, the estimated distribution $M_{pred}$ may encapsulate more information about the map distribution than a mere aggregation of all queries, which can further support query learning.

We incorporate the global information into individual queries as follows: First, we encode the global information representation $M_{pred}$ using a simple MLP:

$$F_{global} = \text{MLP}(\text{Flatten}(M_{pred})), \qquad (8)$$

where $F_{global}$ is regarded as the map global embedding. Then, we incorporate the encoded global information into each individual query by concatenating it with every query. An MLP is then used to fuse the local query with the global

information. The process can be expressed as:

$$q_i = \text{MLP}(\text{concat}([q_i, F_{global}])). \qquad (9)$$

Finally, we employ the above query that incorporates global information to perform the final prediction decoding.

## 4. Experiments

### 4.1. Datasets and Evaluation Metrics

**Datasets** We perform extensive experiments using two publicly available datasets: nuScenes (Caesar et al., 2020) and Argoverse 2 (Wilson et al., 2023). The nuScenes dataset includes 1000 driving scenes collected from Boston and Singapore, each approximately 20 seconds long and consisting of 40 keyframes sampled at 2Hz. In line with previous methods (Liao et al., 2022; 2024; Liu et al., 2024b), we use 700 scenes with 28,130 samples for training and 150 scenes with 6,019 samples for validation. For each sample, the dataset provides 6 perspective images along with corresponding point clouds from a 32-beam LiDAR. Argoverse 2 consists of 1000 scenes collected from six cities, each consisting of 15 seconds of 20Hz RGB images from 7 cameras, 10Hz LiDAR sweeps, and a 3D vectorized map. The dataset is divided into training, validation, and test sets, with 700, 150, and 150 logs, respectively. We mainly focus on three primary categories of map elements: road boundaries, lane dividers, and pedestrian crossings.

**Evaluation Metrics** To ensure a fair comparison with previous methods (Liao et al., 2022; 2024), we adopt Average Precision (AP) as the primary evaluation metric, following prior works, with Chamfer distance as the matching criterion. A prediction is deemed True-Positive (TP) only if its chamfer distance to the ground truth is below a specified threshold. The AP is averaged across two distance thresholds set: 0.2m, 0.5m, 1.0m for $AP_1$ and 0.5m, 1.0m, 1.5m for $AP_2$. The final mean AP (mAP) is calculated by averaging the AP scores across three road element types: pedestrian crossing ($AP_{ped}$), lane divider ($AP_{div}$), and road boundary ($AP_{bou}$). With the ego-car at the center, the perception ranges extending from [-15.0m, 15.0m] along the X-axis and [-30.0m, 30.0m] along the Y-axis.

### 4.2. Implementation Details

The nuScenes dataset provides images at a resolution of $1600 \times 900$. For model training and evaluation, we downscale these images by a factor of 0.5. In the Argoverse 2 dataset, the 7 camera images have varying resolutions: $1550 \times 2048$ for the front view and $2048 \times 1550$ for the remaining views. To standardize the image sizes, we first pad all 7 camera images to 2048 x 2048, then resize them by a factor of 0.3. Additionally, color jitter is applied to both the nuScenes and Argoverse2 datasets by default. We adopt

AdamW (Loshchilov, 2017) optimizer with weight decay 0.01. The default training schedule is 24 epochs, and the initial learning rate is set to $6 \times 10^{-4}$ with cosine decay and cosine annealing schedule.

Unless stated otherwise, we apply the GRL and GRG module on the first two layers of the six-layer decoder. This is because enforcing global distribution constraints conflicts with the objective of final convergence. Specifically, while the global constraint encourages a more dispersed query distribution, effective convergence requires queries to be concentrated around the ground truth. To balance these effects, we limit the application to the first two layers. The weight ratio between the global representation learning loss and the query prediction loss is set at 1.0:0.1.

For the MapTR baseline, following the official settings, we employ 50 instance queries to detect map element instances, with each instance represented by 20 sequential points. The model is trained with 4 NVIDIA V100 GPUs with a batch size of $4 \times 4$. For the MapQR baseline, we follow its settings by employing 100 instance queries to detect map element instances. Our model is trained on 4 NVIDIA A800 GPUs using a batch size of $4 \times 8$. We use ResNet50 (He et al., 2016) in all experiments as the backbone for comparison.

### 4.3. Comparisons with State-of-the-art Methods

**Results on nuScenes.** We conduct experiments on multiple baselines to assess the effectiveness of our proposed query global distribution learning strategy for online HD map construction. To ensure a fair comparison with previous work, we select three baselines: MapTR (Liao et al., 2022), MapTRv2 (Liao et al., 2024), MapQR (Liu et al., 2024b), and follow their official experiments setting unless otherwise specified. As shown in Table 1, our method consistently enhances performance across all baselines when integrated as a plug-in module. With MapTR as the baseline, our method improves mAP$_1$ by 4.2% and mAP$_2$ by over 4.9% after training for 110 epochs, while yielding gains of 3.9% in mAP$_1$ and 4.2% in mAP$_2$ after 24 epochs. For MapTRv2, our method achieves a 3.9% improvement in mAP$_1$ and 3.5% improvement in mAP$_2$ after 24 epochs, with corresponding gains of 2.5% and 2.4% after 110 epochs. When using MapQR as the baseline, our approach achieves state-of-the-art performance, achieving a substantial 3.2% improvement in mAP$_1$ and 2.8% in mAP$_2$ after training for 24 epochs. It ultimately reaches mAP$_1$ of 45.3% and mAP$_2$ of 68.1% on the nuScenes validation set.

**Results on Argoverse 2.** In Table 2, we summarize the experimental results on Argoverse 2. All models were trained for 6 epochs with ResNet-50 as the backbone while keeping the dimensions consistent with baselines. Our proposed methods consistently outperform the tested base-

*Table 1.* Comparison with state-of-the-art methods on nuScenes validation set. The best results are highlighted in bold. Grey indicates the reproduced result in our setting, the rest APs are taken from the papers. "-" means that the corresponding results are not available.

| Method | Epoch | $AP_{div}$ | $AP_{ped}$ | $AP_{bou}$ | $mAP_1$ | $AP_{div}$ | $AP_{ped}$ | $AP_{bou}$ | $mAP_2$ |
|---|---|---|---|---|---|---|---|---|---|
| BeMapNet (Qiao et al., 2023) | 30 | 46.9 | 39.0 | 37.8 | 41.3 | 62.3 | 57.7 | 59.4 | 59.8 |
| | 110 | 52.7 | 44.5 | 44.2 | 47.1 | 66.7 | 62.6 | 65.1 | 64.8 |
| StreamMapNet (Yuan et al., 2024) | 24 | 42.9 | 32.3 | 33.2 | 36.2 | 64.1 | 58.2 | 59.4 | 60.6 |
| PivotNet (Ding et al., 2023) | 24 | 41.4 | 34.3 | 39.8 | 38.5 | 56.5 | 56.2 | 60.1 | 57.6 |
| MapTR (Liao et al.) | 24 | 30.7 | 23.2 | 28.2 | 27.3 | 51.5 | 46.3 | 53.1 | 50.3 |
| | 110 | 40.5 | 31.4 | 35.5 | 35.8 | 59.8 | 56.2 | 60.1 | 58.7 |
| **MapTR + Ours** | 24 | **36.2** | **25.9** | **31.6** | **31.2**(+3.9↑) | **57.1** | **50.1** | **56.4** | **54.5**(+4.2↑) |
| | 110 | **45.6** | **34.8** | **39.4** | **40.0**(+4.2↑) | **65.5** | **60.0** | **65.1** | **63.6**(+4.9↑) |
| MapTRv2 (Liao et al., 2024) | 24 | 40.0 | 35.4 | 36.3 | 37.2 | 62.4 | 59.8 | 62.4 | 61.5 |
| | 110 | 49.0 | 43.6 | 43.7 | 45.4 | 68.3 | 68.1 | 69.7 | 68.7 |
| **MapTRv2 + Ours** | 24 | **44.0** | **39.1** | **40.3** | **41.1**(+3.9↑) | **64.9** | **64.5** | **65.6** | **65.0**(+3.5↑) |
| | 110 | **52.3** | **45.8** | **45.6** | **47.9**(+2.5↑) | **71.3** | **69.8** | **72.3** | **71.1**(+2.4↑) |
| MapQR (Liu et al., 2024b) | 24 | 49.9 | 38.6 | 41.5 | 43.3 | 68.0 | 63.4 | 67.7 | 66.4 |
| | 110 | 57.3 | 46.2 | 48.1 | 50.5 | 74.4 | 70.1 | 73.2 | 72.6 |
| MapQR | 24 | 48.1 | 36.9 | 41.2 | 42.1 | 66.8 | 61.9 | 67.1 | 65.3 |
| **MapQR + Ours** | 24 | **51.6** | **41.3** | **42.9** | **45.3**(+3.2↑) | **69.8** | **65.8** | **68.8** | **68.1**(+2.8↑) |
| | 110 | **58.9** | **48.1** | **48.8** | **51.9**(+1.4↑) | **75.3** | **70.5** | **73.6** | **73.1**(+0.5↑) |

*Table 2.* Comparison with SOTA methods on Argoverse 2. Grey indicates the reproduced result in our setting.

| Method | $AP_{div}$ | $AP_{ped}$ | $AP_{bou}$ | $mAP_1$ | $AP_{div}$ | $AP_{ped}$ | $AP_{bou}$ | $mAP_2$ |
|---|---|---|---|---|---|---|---|---|
| MapTR | 40.5 | 27.9 | 32.9 | 33.7 | 58.7 | 55.4 | 59.1 | 57.8 |
| **MapTR + Ours** | 41.4 | 29.6 | 34.8 | 35.3(+1.6↑) | 58.9 | 57.3 | 60.4 | 58.9(+1.1↑) |
| MapTRv2 | 48.1 | 30.5 | 36.7 | 38.4 | 69.1 | 59.8 | 65.3 | 64.7 |
| **MapTRv2 + Ours** | 49.0 | 32.1 | 37.8 | 39.6(+1.2↑) | 69.5 | 61.6 | 65.8 | 65.6(+0.9↑) |
| MapQR | 53.6 | 33.4 | 39.4 | 42.1 | 69.8 | 60.5 | 65.0 | 65.1 |
| **MapQR + Ours** | 53.1 | 34.6 | 41.0 | 42.9(+0.8↑) | 69.8 | 61.8 | 66.4 | 66.0(+0.9↑) |

*Table 3.* Ablation study of different components. Numbers in parentheses indicate improvement over the baseline setting.

| MapTRv2 | GRL | GRG | $AP_{div}$ | $AP_{ped}$ | $AP_{bou}$ | $mAP_2$ |
|---|---|---|---|---|---|---|
| ✓ | | | 59.8 | 62.4 | 62.4 | 61.5 |
| ✓ | ✓ | | 63.8 | 63.0 | 64.2 | 63.7 (+2.2↑) |
| ✓ | ✓ | ✓ | 64.9 | 64.5 | 65.6 | 65.0 (+3.5↑) |

the first and third rows, our results are more accurate and visually smoother in the mini-roundabout region, better capturing the road environment. This improvement is particularly evident in the central triangle region of the first row. Additional visual results comparison are provided in the supplementary material.

### 4.5. Ablations

**Components Ablation:** To further demonstrate the effectiveness of the two components in our method, we conduct an ablation study on GRL and GRG modules, as presented in Table 3. The first row represents the original MapTRv2, which does not incorporate any global information. In the second row, we introduce global representation learning for the queries. The third row, which integrates both components, results in further performance improvements.

**Ablation on Feature Dimensions in GRL:** We analyze the impact of the MLP dimension in Equation 4 on perfor-

lines. Notably, the MapQR-based baseline achieves the highest performance, reaching $mAP_1$ of 42.9% and $mAP_2$ of 66.0%. Furthermore, compared to MapTR and MapTRv2, our method demonstrates an approximate 1% performance gain.

### 4.4. Visualizations

Qualitative results in Figure 8 demonstrate that our method consistently improves map prediction across various driving scenarios compared to previous methods. Notably, in

*Table 4.* Ablation study on the feature dimension of GRL module.

| Dimension | 512 | 2048 | 8192 |
|---|---|---|---|
| $mAP_2$ | 64.6 | 65.0 | 64.2 |

*Table 5.* Ablation study (on MapQR baseline) on the effect of applying our methods to the first $N$ layers of the decoder.

| $N$ | $AP_{div}$ | $AP_{ped}$ | $AP_{bou}$ | $mAP_2$ |
|---|---|---|---|---|
| 1 | 68.36 | 64.13 | 68.61 | 67.03 |
| 2 | 69.78 | 65.75 | 68.84 | 68.12 |
| 3 | 68.75 | 63.81 | 68.78 | 67.11 |

mance, the results are shown in Table 4. It can be seen that performance drops at low dimensions due to information loss, and also drops at high dimensions due to overfitting caused by increased parameters, which limits generalization.

**Auxiliary Layers** Ablation study on $N$, where the first $N$ transformer layers are constrained. Our methods were applied to queries from layer $N$ like below:

$$\mathbf{Q}_{i+1} = \begin{cases} \mathcal{C}(\mathbf{Q}_i^{output}), & i \leq N \\ \mathbf{Q}_i^{output}, & i > N, \end{cases} \quad (10)$$

where $\mathcal{C}$ denotes the applying of our methods, $\mathbf{Q}_i^{output}$ denotes the Query output from layer $i$. In this ablation experiment, we apply our method to the query outputs of the first $N$ layers of the decoder transformer while keeping the remaining $(6 - N)$ layers unchanged. As shown in Table 5, our method get best performance when applied on the first two layers.

**Sensitivity to the Loss Weight** We sweep the ratio between the GRL loss $\mathcal{L}_{global}$ and the query prediction loss on MapQR + MapGR in Table 6. Across an order of magnitude of variation, mAP changes by less than $0.5$, indicating that the default $1.0 : 0.1$ setting is not a finely tuned hyperparameter and does not require dataset-specific search.

**Disentangling Global Representation Learning from Auxiliary Supervision**

Both MapTRv2 and MapQR already employ perspective-view (PV) and BEV segmentation supervision. To verify that MapGR's gains stem from *query-side global representation learning* rather than from additional supervision, we conduct the ablation in Table 7 on MapQR.

Three observations support that the gain is attributable to global representation learning rather than extra supervision: The standalone GRG module ($+1.6$ mAP over baseline, row 2) introduces no new loss signal; it only re-injects an aggregated query embedding. Replacing MapQR's BEV-feature segmentation head with our query-driven GRL head

*Table 6.* Sensitivity of MapGR to the GRL auxiliary loss weight ($\mathcal{L}_{global} : \mathcal{L}_{query}$), MapQR baseline on nuScenes *val*.

| Loss ratio | $1.0 : 0.05$ | $1.0 : 0.1$ | $1.0 : 0.2$ |
|---|---|---|---|
| mAP | 67.7 | **68.1** | 67.6 |

*Table 7.* Disentangling the contribution of MapGR from auxiliary BEV segmentation on MapQR (nuScenes *val*). "PV + BEV seg" is the baseline's existing segmentation head; "GRL seg" is our query-aggregated rasterized prediction.

| Method | Seg. Supervision | mAP |
|---|---|---|
| MapQR (baseline) | PV + BEV seg | 65.3 |
| MapQR + Ours | PV + BEV seg | 66.9 |
| MapQR + GRL only | PV + GRL seg | 67.5 |
| MapQR + Ours | PV + BEV seg + GRL seg | **68.1** |

under an identical supervision budget yields $+2.2$ mAP (row 3), showing that the same BCE signal is more effective when it shapes the query output distribution than when it shapes the BEV feature map.

### 4.6. Efficiency analyses

In autonomous driving tasks, resources are highly limited, thus imposing stringent efficiency requirements on the model. Consequently, we analyze the impact of introducing the GRL and GRG modules proposed in this paper on the model's efficiency. The results are shown in Table 8. The added GRL and GRG module increases the parameters due to MLP, but adds negligible overhead, only brings parameter growth (4–23% depending on baseline) and 1% overhead ($\leq 1.2$ ms).

*Table 8.* Efficiency comparison with baselines.

| Methods | Parameter | FPS |
|---|---|---|
| MapTR | 47.5M | 24.2 |
| **MapTR + Ours** | **58.5M** | **23.8** |
| MapTRV2 | 56.1M | 19.6 |
| **MapTRV2 + Our**s | **67.1M** | **19.4** |
| MapQR | 225.4M | 17.9 |
| **MapQR + Ours** | **236.4M** | **17.7** |

## 5. Discussion

**Explanation from Another Perspective** Our work can be regarded as a method of incorporating map structures as prior information into the query. By embedding global information within the query, we enhance the relative interactions between queries, making it easier to learn associations between targets. Different from independent object detection, these associations are reflected in maps; for instance, adjacent lane markings typically exhibit smooth curvature transitions, while neighboring road elements tend to align

*Table 9.* Re-enabling GRL+GRG at L2 under higher mask resolution. Baseline: MapQR on nuScenes *val*.

| Setting | Mask resolution | $mAP_2$ |
|---|---|---|
| Ours (L0–L1) | $200{\times}100$ | 68.1 |
| + L2 at $4\times$ res. | $800{\times}400$ | **68.5** |

*Table 10.* Downstream motion forecasting with HiVT, consuming online HD maps from MapTRv2 with and without MapGR.

| Online HD map source | minADE↓ | minFDE↓ | MR↓ |
|---|---|---|---|
| MapTRv2 | 0.4057 | 0.8499 | 0.0992 |
| MapTRv2 + MapGR | **0.3822** | **0.7920** | **0.0817** |

at endpoints and maintain similar lengths. Additionally, visualization results indicate that our method produces better outputs, further validating these structural properties.

**Can a Simple MLP Encode the Whole Map Properly?**
In our experiments, the rasterized HD map is encoded into a 256-dimensional feature vector via an MLP, which we deem sufficient to capture its key information. To validate this, we conducted a reconstruction experiment (a) on the nuScenes dataset. We randomly sampled 2,000 rasterized maps from the training subset of the nuScenes dataset for training, and 200 maps from the validation subset for evaluation. Reconstruction results on val set are shown as Figure 4.

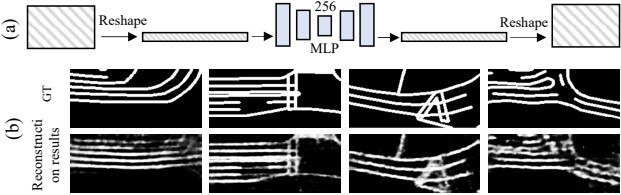

*Figure 4.* Results of MLP-based encoding and decoding.

**Why Inject Global Guidance Only at the Early Layers?**
The default GRL mask has resolution $200{\times}100$ over a $60\,\text{m}\times 30\,\text{m}$ BEV range, so a single cell covers $0.3\,\text{m}$, or about $0.01$ in normalized $y$-coordinates. Tracking the per-layer query displacement on the trained decoder (Supplementary Fig. 6) shows two clearly distinct regimes: The cross-layer MAE variance of identical queries from L0 to L1 is at least $0.02$, larger than one mask cell; from L2 onward the MAE drops below $0.01$, smaller than a cell. Once the per-layer query update is finer than the mask resolution, the global constraint can no longer provide finer-grained supervision than the local per-instance refinement already taking place, and the two signals begin to conflict.

To verify that the layer choice is a resolution effect rather than a hand-tuned hyper-parameter, we quadruple the mask resolution to $800\times400$ (cell size $0.075\,\text{m}$) and re-enable GRL + GRG at layer 2. As shown in Table 9, the L2 gain

*Table 11.* Effect of training schedule on Argoverse 2 (mAP). Baseline: MapQR.

| Training schedule | Base | Ours |
|---|---|---|
| 6 epochs | 65.1 | 66.0 (+0.9) |
| 6 + 6 epochs | 65.7 | **67.6** (+1.9) |

is recovered once the supervision precision surpasses the inter-layer query motion. We retain the $200{\times}100$ resolution and the first-two-layer setting in the main results because the $4\times$ resolution raises the GRL projection-head memory and compute cost by roughly $16\times$, which is incompatible with real-time requirements.

**Impact on Downstream Motion Forecasting Task** To investigate whether the performance improvements brought by MapGR can yield benefits for downstream driving tasks, we feed the predicted online HD maps into the HiVT (Zhou et al., 2022) motion forecasting model and evaluate on nuScenes *val*. The experimental results are shown in Table 10. It can be observed that all three forecasting metrics are consistently improved, with a $-5.79\,\text{cm}$ reduction in minFDE and a $-1.75\%$ absolute drop in miss rate. The gains confirm that higher-quality vectorized maps produced by MapGR translate into measurable benefits for a downstream task that was not co-trained with this module.

**Does Long-Schedule Training Benefit Argoverse 2 Performance?** To probe whether the Argoverse 2 gains saturate under the standard short schedule, we extend training to a 6-epoch pre-training plus 6-epoch fine-tuning regime, using MapQR as the baseline, and experiment results are shown in Table 11. The improvement grows from $+0.9$ to $+1.9\,\text{mAP}$, indicating that the benefit is *not* a short-schedule artifact and continues to widen as the baseline converges.

# 6. Conclusion

In this paper, we propose to leverage the global representation learning from queries to enhance the quality of map perception. Our method includes a Global Representation Learning (GRL) module, which aims to learn a global representation from all queries, and a Global Representation Guidance (GRG) module, which utilizes global information to guide the optimization process of local queries. The proposed approach functions as a plug-and-play module that can be seamlessly integrated into most mainstream methods. Our experimental results demonstrate that incorporating our method significantly enhances performance without increasing computational cost, and achieves state-of-the-art results on both the nuScenes and Argoverse 2 datasets. Future research could focus on developing more effective global representations and supervision strategies for map to further enhance query learning.

## Impact Statement

This paper presents a method for online vectorized HD map construction, with the goal of improving machine learning systems for autonomous driving perception. Potential positive impacts include improved map quality and more reliable downstream driving-related tasks. Potential risks include misuse or over-reliance in safety-critical autonomous systems without sufficient validation under diverse real-world conditions. We emphasize that deployment of such systems should require rigorous safety evaluation, robustness testing, and compliance with applicable traffic, privacy, and safety regulations.

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

# A. Appendix

## A.1. Full Details and Additional Results

**Additional Ablation on Loss Weights** Ablations about the weight of original detection loss on affected layers are presented in Table 12. $\omega = 0.1$ corresponds to appropriate supervision and is adopted as the default setting.

*Table 12.* Ablation on layer loss weights

| $\omega$ | $AP_{ped}$ | $AP_{div}$ | $AP_{bou}$ | $mAP_2$ |
|---|---|---|---|---|
| 0.1 | 67.88 | 63.82 | 67.88 | 66.53 |
| 0.3 | 67.07 | 61.42 | 67.46 | 65.32 |

**Gradient Weakening** In scenarios where a conflict occurs between global distribution learning and map-based object detection—such as encountering gradient explosion—we mitigate the issue by applying gradient weakening to the processed query. Specifically, we modify the gradient computation as follows: $f(X) = X \cdot (1 - \theta) + X_{\text{detach}} \cdot \theta$. The Ablation of the $\theta$ are presented in Table 13.

*Table 13.* Ablation study of $\theta$ used in gradient weakening

| $\theta$ | $AP_{div}$ | $AP_{ped}$ | $AP_{bou}$ | $mAP_2$ |
|---|---|---|---|---|
| 0.9 | 67.88 | 63.82 | 67.88 | 66.53 |
| 0.8 | 69.21 | 63.46 | 67.83 | 66.83 |
| 0.7 | 67.48 | 63.14 | 66.05 | 65.56 |

**More Ablation of How Global Features are Generated** We compare three approaches for global feature generation: BEV Segmentation Mask (BSM), Cross Attention (CA) in Figure 5, and our proposed GIG method. Evaluated on the nuScenes validation set as shown in Table 14, GIG demonstrates superior performance:

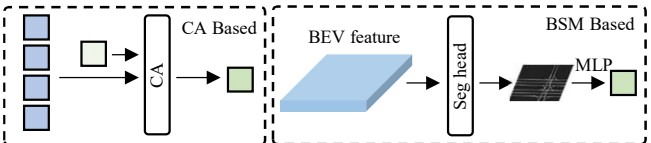

*Figure 5.* Details of different global information generation approaches.

*Table 14.* Comparison of using different global representation learning methods

| Baseline (w/o global embedding) | BSM Based | CA Based | GIG (Ours) |
|---|---|---|---|
| 61.5 | 62.1 | 63.6 | 65.0 |

**Cross-Layer Query Stability Analysis** By tracking the coordinates of all queries during the inference process, we statistically analyze the fluctuation and stability of queries across Transformer layers in MapQR, both with and without our proposed method. We use the mean average error (MAE) of query coordinates changes across layers as the metric to reflect its stability. As illustrated in the comparison in Figure 6, the application of our method to the first two layers results in significantly lower "volatility" in the subsequent four layers. This suggests that the introduction of global queries fosters a more effective initial query distribution within the early layers, thereby reducing subsequent query variations. Furthermore, Figure 7 demonstrates that our method leads to higher overall inter-layer stability of the queries.

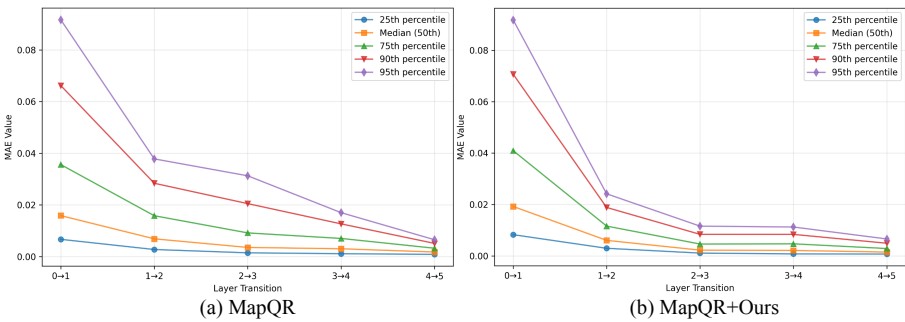

(a) MapQR        (b) MapQR+Ours

*Figure 6.* A cross-layer-wise comparison of query stability reveals that with our method, queries become significantly more stable after the first two stages (0, 1).

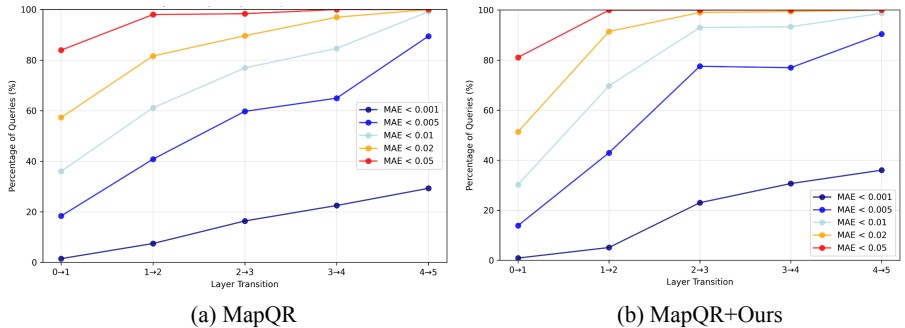

(a) MapQR        (b) MapQR+Ours

*Figure 7.* Overall query stability comparison across layers.

**Detail Hyperparameters** Below Table 15 is the detailed list of hyperparameters adopted for training MapGR on the nuScenes dataset.

*Table 15.* Hyperparameters used for training MapGR on nuScenes dataset.

| Hyperparameter | Value |
| --- | --- |
| Learning Rate | 6e-4 |
| Batch Size | 4 x 8 |
| Optimizer | AdamW |
| Weight Decay | 0.01 |
| Learning Rate Scheduler | Cosine Annealing |
| Warm-up Steps | 500 |
| Number of Epochs | 24 or 110 |
| Dropout Rate | 0.1 |
| Number of Queries | 100 |
| MapGR Applied Layers | 2 |
| Loss Function of GRL | BCE |
| Loss Ratio on Applied Layers | 1.0 |
| Ratio of Other Loss on Applied Layers | 0.2 |
| Number of Segmentation Classes | 3 |
| Gradient Weakening Coefficient | 0.8 |

**Additional Visualization** Figure 8 and Figure 9 present a visual comparison between MapQR augmented with our method, the original MapQR baseline, and MapTRv2.

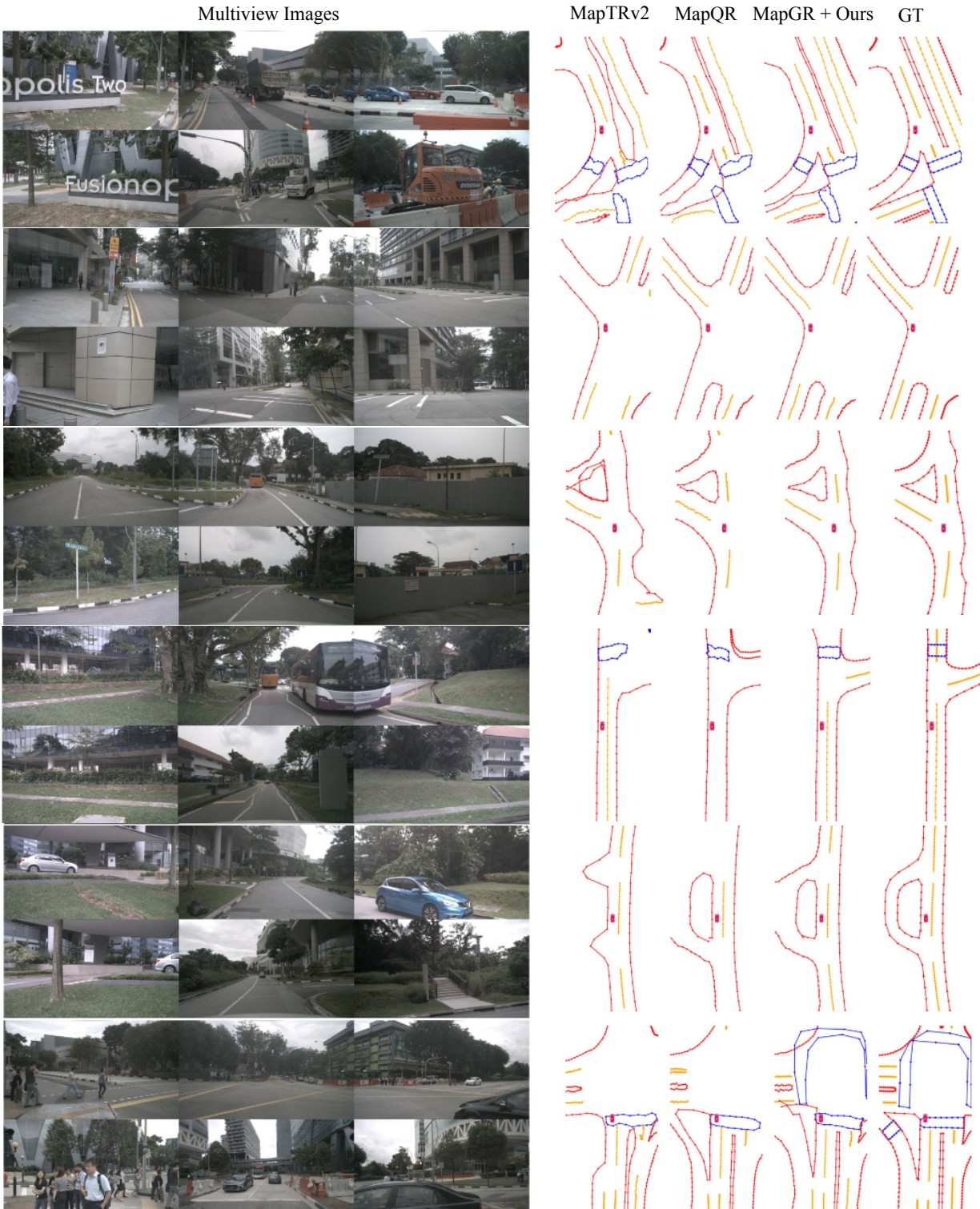

*Figure 8.* Qualitative comparison between our methods with MapQR and MapTRv2 on the nuScenes validation dataset.

Multiview Images           MapTRv2   MapQR  MapQR + Ours  GT

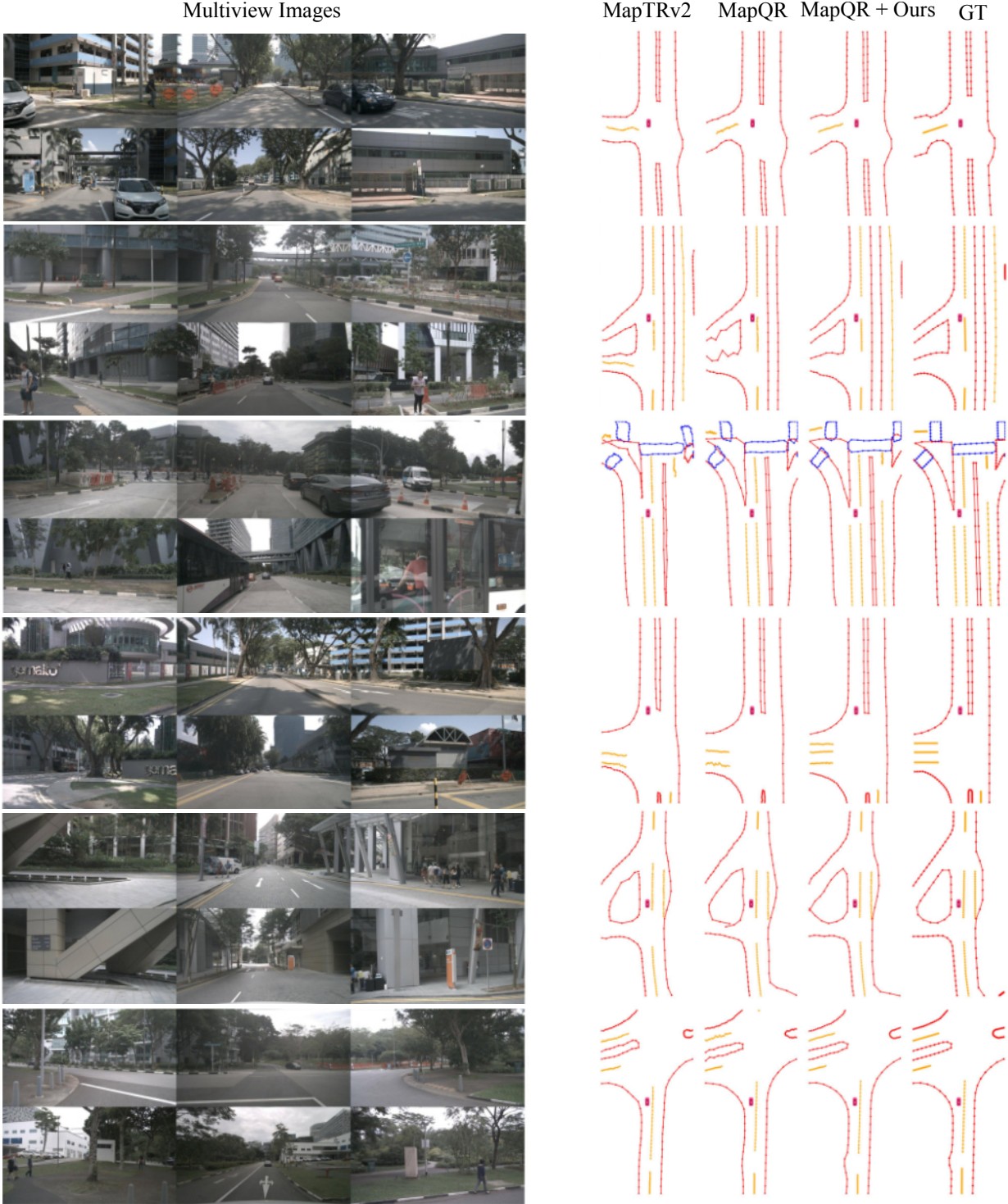

*Figure 9.* Additional qualitative comparison between our methods with MapQR and MapTRv2 on nuScenes dataset.

