# OpenReview forum: "Learning Global Representation from Queries for Vectorized HD Map Construction"
_ICML.cc/2026/Conference — ICML 2026 regular_

### Official Review · Reviewer_mQGL · 2026-03-09

**Soundness:** 3
**Presentation:** 2
**Significance:** 3
**Originality:** 2
**Overall Recommendation:** 4
**Confidence:** 3

**Summary:**

This paper proposes MapGR, a global representation learning approach for HD Map construction. In MapGR, a Global Representation Learning (GRL) module and a Global Representation Guidance (GRG) module are designed. This paradigm learns a global representation and utilizes this to guide the optimization process of local queries, and finally refined the vectorized HD map construction. Extensive experiments are conducted that the incorporating of this plug-and-play module enhances the performance without increasing computational burden.

**Compliance With Llm Reviewing Policy:**

Affirmed.

**Final Justification:**

My Concerns are fully resolved. I choose to raise my score to 4.

**Key Questions For Authors:**

See the part of weakness.

**Limitations:**

No.

For limitations and potential negative societal impact, the consideration on ensuring the real-deployed inference speed and model generalization ability are reuiqred.

**Strengths And Weaknesses:**

Strengths:

1.	This paper is well-written, and the expression is clear. Figures and Tables are clear to express the MapGR design and effectiveness.

2.	Global Representations are essential for map constructions, especially when there are obstructions or low-light scenarios. The motivation is reasonable.

3.	Using segmentation loss to construct the global representative is reasonable and effective.

4.	Experiments are extensive to show the effectiveness based on different HD mapping baselines.

Weakness:

1.	In Figure1(c), it is hard to convince that the map's global embedding assists in the improvement. More explanations are required for the global embedding.

2.	Lack of novelty. The effectiveness of using segmentation loss has been proposed in MGMap [1]. More distinctions and comparative experiments are required to show the effectiveness and specific design.

3.	It is doubtful that the model’s benefits may partly come from additional supervision, not entirely from the global representation itself. This point requires more specification.

4.	Grammar mistakes: line 53-54: “aims to learns” line 86-87: “which can guides”. In experiments: “the results is”.


[1] Liu, X., Wang, S., Li, W., Yang, R., Chen, J., & Zhu, J. (2024). Mgmap: Mask-guided learning for online vectorized hd map construction. In Proceedings of the IEEE/CVF Conference on Computer Vision and Pattern Recognition (pp. 14812-14821).

---

> ### Author Rebuttal · Authors · 2026-03-31
>
> **Unclear Motivation for Global Embedding:**
>
> Figure 1(c) illustrates that incorporating global embedding progressively aligns query distribution from initial to final decoder layers, resulting in smoother and more consistent predictions (highlighted by the red box). This visual evidence, combined with the quantitative gains in Table 3 in manuscript clearly indicate that the GRG module provides a substantial performance boost. Furthermore, in Table 9 of the Supplementary Material, we investigated the impact of different global embedding generation strategies on the final performance. It is evident that our method yields consistent and significant improvements across all evaluated configurations, further validating the robustness of our design. We believe that this improvement stems from the fact that the global embedding effectively contextualizes individual queries with global scene-level information. By incorporating insights from the entire query set, our module facilitates more informed and robust query learning in the subsequent decoder stages.
>
> **Novelty vs. MGMap:**
>
> We thank the reviewer for this comparison. While both methods leverage rasterized map supervision, they differ fundamentally in design, efficiency, and performance:
>
> | Aspect | MGMap| MapGR (Ours) |
> |---|---|---|
> | Segmentation role | Primary guidance per instance (local) | Auxiliary supervision for global structure |
> | Query interaction | Queries remain independent | All queries jointly aggregated |
> | Global feedback | None | Global embedding $\rightarrow$ each query (GRG) |
> | FPS overhead |  <11.6 | 23.8 |
> | mAP$_2$ (MapTRv2 based, 24ep) | 64.8 | 65.0 |
>
> Our method distinguishes itself from existing segmentation-based approaches by generating predictions directly from instance queries. This query-driven mechanism imposes distributional constraints to facilitate representation learning, further enhanced by re-injecting global context. Notably, our approach surpasses MGMap in accuracy while requiring significantly lower computational overhead, demonstrating its superior practical efficiency.
>
> **Performance Gain Attribution:**
>
> We clarify that part of the performance gains come from our global representation learning, not merely from additional supervision. Notably, both MapTRv2 and MapQR already incorporate perspective-view (PV) and BEV segmentation supervision in their original frameworks — our method does not introduce any extra supervision signals beyond what the baselines already use.
>
> To further disentangle the two factors, we conduct the following ablation on MapQR:
> | Method | Seg. Supervision | mAP |
> |---|---|---|
> | MapQR (baseline) | PV + BEV seg | 65.3 |
> | MapQR + Ours (w/o any seg supervision) | PV + BEV seg | 66.9 |
> | MapQR w/ GRL only (BEV seg transferred) |  PV + GRL seg | 67.5 |
> | MapQR + Ours (full) |  PV + BEV seg + GRL seg | 68.1 |
>
>   Several observations support that the gain is attributed to global representation learning:
> - Supervision is not increased. MapQR already uses BEV seg supervision; our GRL module simply use the same seg supervision as global representation learning signal rather than adding new supervision.
> - Global representation alone improves performance. Even without any segmentation supervision, MapQR + Ours achieves 66.9 mAP, already outperforming the baseline (65.3), demonstrating the intrinsic value of global representation guidance (GRG).
> - Transferring BEV seg to GRL still helps. Replacing MapQR's original BEV seg with our GRL-based seg yields 67.5 mAP, confirming that the structured global learning signal in GRL is more effective than standalone BEV supervision.
> - Full method achieves best result (68.1 mAP), combining both global representation learning and guidance.
>
>
> **Grammar mistakes:**
> We appreciate the reviewer’s meticulous feedback regarding the writing. We will carefully polish the entire paper, meticulously addressing any other latent grammatical errors and linguistic inconsistencies, to ensure the highest quality for the final version.
>
> **Inference speed and generalization ability:**
> We provide a detailed efficiency analysis in Table 6. The results demonstrate that our approach introduces minimal impact on real-time performance. Regarding the generalizability of our approach, we conducted extensive experiments across three different baselines and two challenging public datasets. The empirical results consistently demonstrate that our method yields significant performance gains across various settings, validating its robustness and versatility.

---

> > ### Author Rebuttal · Reviewer_mQGL · 2026-04-03
> >
> > Thanks for the authors' response. My concerns are fully resolved. I choose to raise my score to 4.

---

> > > ### Author Response · Authors · 2026-04-03
> > >
> > > Thank you sincerely for taking the time to carefully read our rebuttal throughout the review process. We are glad that our responses have adequately addressed your concerns. We will incorporate all the suggested clarifications and corrections (including the grammar fixes and additional ablation details) into the final version of the manuscript. We truly appreciate your thoughtful engagement and your willingness to update your assessment.

---

### Official Review · Reviewer_GBF9 · 2026-03-11

**Soundness:** 2
**Presentation:** 3
**Significance:** 2
**Originality:** 3
**Overall Recommendation:** 4
**Confidence:** 2

**Summary:**

This paper aims to addressing the flaw in current HD Map construction, as existing models treat map elements as "spiky" independent objects, whereas maps actually have "streak" distributions (continuous lines). MapGR fixes this by learning a global representation.
Experiments show strong gain and plug-and-play versatility on top of different frameworks.

**Compliance With Llm Reviewing Policy:**

Affirmed.

**Final Justification:**

My main concerns are resolved. Thus, I keep my positive score 4.

**Key Questions For Authors:**

see weekness

**Limitations:**

This paper does not include mandatory impact statement.

**Strengths And Weaknesses:**

Strength:
1. Addresses a valid limitation: It identifies that existing models treat map elements as "spiky" independent objects, whereas maps actually have "streak" distributions (continuous lines). MapGR fixes this by learning a global representation.
2. Plug-and-Play Versatility: One of the paper's strongest points is that its modules are designed to be integrated into existing frameworks like MapTR and MapQR without requiring a total architectural overhaul.
3. Solid Empirical Gains: The method shows consistent improvements in mean Average Precision (mAP) across two major datasets: nuScenes and Argoverse 2.

Weakness:
1. My main concern is if the improved HD map would in general help with down streaming task of driving. Figure 1 gives one example that supports the impact of improved map. But most examples in Figure 9 seem to show that the driving decision would not change much. The gain may not sufficiently justify the overhead, though I agree that the overhead is small.
2. In the experiment part, the weight ratio between the global representation learning loss and the query prediction loss is set at 1.0:0.1.  And in the ablation study part, only one additional number 0.3 is tested. This may not sufficient address question in hyper-parameter sensitivity:  if moved to a new dataset, would the current number 0.1 be sufficient? or would the user need to do hyper-parameter search?

---

> ### Author Rebuttal · Authors · 2026-03-31
>
> **Downstream Task Utility:**
>
> To evaluate the efficacy of our method in downstream applications, we integrated our output into the HiVT model. The experimental results, as summarized below, demonstrate that our approach consistently facilitates significant performance gains for downstream tasks. Regarding the visualizations, while the improvements in certain cases in Figure 9 may not significant relative to the baseline, as we want to provide a comprehensive and faithful evaluation, so we have included samples where the performance gains are less pronounced, the results presented in Figure 8 exhibit substantial qualitative enhancements. our empirical evaluations also demonstrate that these improvement provide substantial benefits to downstream tasks.
>
> | Online HD Map Method | minADE ↓ | minFDE ↓ | MR ↓ |
> |---|---|---|---|
> | MapTRv2 | 0.4057 | 0.8499 | 0.0992 |
> | **MapTRv2 + Ours** | **0.3822** | **0.7920** | **0.0817** |
>
> Our MapGR consistently improves all three motion forecasting metrics, with notable reductions in minFDE (−5.7 cm) and miss rate (−1.75%). The downstream gains confirm that MapGR's higher-quality map predictions translate to meaningful benefits for real-world driving applications.
>
> **Hyper-parameter Sensitivity of Loss Weights:**
>
> We appreciate this concern and conducted additional ablation studies on MapQR+Ours to evaluate the sensitivity of the loss weight ratio. The results are as follows:
>
> | Loss Ratio | 1.0:0.05 | 1.0:0.1 | 1.0:0.2  |
> |---|---|---|---|
> | mAP | 67.7 | 68.1 | 67.6 |
>
> The performance remains stable across a wide range of weight ratios (67.6–68.1 mAP), with a variation of less than 0.5 mAP. This demonstrates that MapGR is not sensitive to this hyper-parameter. The default ratio of 1.0:0.1 performs best, but neighboring values yield comparable results, suggesting that practitioners applying MapGR would not require an extensive hyper-parameter search.

---

> > ### Author Rebuttal · Reviewer_GBF9 · 2026-04-03
> >
> > Thanks for the rebuttal. My concern is resolved.

---

> > > ### Author Response · Authors · 2026-04-08
> > >
> > > Thank you sincerely for taking the time to carefully read our rebuttal throughout the review process. We are glad that our responses have adequately addressed your concerns.

---

### Official Review · Reviewer_Fz97 · 2026-03-13

**Soundness:** 3
**Presentation:** 3
**Significance:** 3
**Originality:** 2
**Overall Recommendation:** 4
**Confidence:** 4

**Summary:**

This paper studies camera-based vectorized HD map construction and argues that existing DETR-style query-based methods over-emphasize instance-level local optimization while underusing global map context. To address this, the paper proposes MapGR, a plug-in module with two components: GRL, which aggregates all queries into a holistic rasterized map prediction and applies BCE supervision, and GRG, which feeds the resulting global representation back into the queries. The method can be integrated into existing query-based frameworks and shows consistent gains on nuScenes and Argoverse 2 with MapTR, MapTRv2, and MapQR.

**Compliance With Llm Reviewing Policy:**

Affirmed.

**Final Justification:**

MapGR is a simple, modular plug-in that consistently improves multiple baselines on two benchmarks with negligible overhead. The rebuttal strengthened the paper by providing additional structural metrics and a principled resolution-based justification for the layer selection design. Originality remains modest as the individual components are well known, though their application directly at the decoder query stage is a reasonable design contribution. The rebuttal adequately addressed my concerns and reinforced my prior assessment. I maintain my score of 4.

**Key Questions For Authors:**

- Can the paper include metrics that directly measure structural quality, such as connectivity, fragmentation, or endpoint consistency?

- How much of the gain comes from the specific GRG design, versus simply adding dense global auxiliary supervision?

- Why is applying MapGR to the first two decoder layers the best design choice?

- Do the gains on Argoverse 2 remain under longer training schedules?

**Limitations:**

yes

**Strengths And Weaknesses:**

Strengths

- The proposed method is simple, modular, and easy to add to existing frameworks.

- The empirical gains are consistent across two benchmarks and multiple strong baselines.

- The ablations are reasonably thorough, covering GRL/GRG, insertion depth, representation dimension, and efficiency.

- The paper is clearly written and easy to follow.

Weaknesses

- The novelty is somewhat limited. It mainly combines global auxiliary raster supervision with query guidance.

- The claim about learning “global structural information” is somewhat overstated, since the supervision is still raster-mask BCE rather than explicit topology modeling.

- The evaluation does not directly measure structural quality such as connectivity or fragmentation.

- The mechanism is not fully isolated; stronger comparisons against simpler dense auxiliary supervision baselines would help.

- Some design choices, such as applying the module only to the first two decoder layers, are not deeply analyzed.

---

> ### Author Rebuttal · Authors · 2026-03-31
>
> **Method novelty**:
>
> The core strength of our approach lies in its unique ability to impose global distributional constraints on the collective query space by supervising a holistic rasterized map prediction, and subsequently feed the learned global embedding back into each query via GRG. By integrating global contextual information into individual queries, our approach provides explicit guidance for query learning. Empirical results confirm that our framework consistently outperforms both the baseline and state-of-the-art methods, such as MapTRv2 and MGMap, which utilize dense auxiliary supervision or mask guidance."
>
> **"Global structural information" statement:**
>
> We sincerely appreciate the reviewer’s insightful suggestion. Upon careful consideration, we agree that the term 'structural' may indeed be somewhat overstated in this context, and 'global information' to be a more accurate and rigorous description. We have identified three instances of this terminology throughout the manuscript and will revise them accordingly in the final version to ensure terminological precision.
>
> **Structural quality measurement:**
>
>  We evaluate three additional structural metrics on the nuScenes validation set beyond mAP with the MapTR as baseline:
>
> 1. Rasterized IoU (RIoU): Buffers each predicted/GT polyline by 1.0m and computes pixel-level overlap. Top-N filtering (N = GT instance count per class) isolates structural coverage from over-detection artifacts.
>
> | Method | Divider | Ped. Crossing | Boundary | Mean |
> |---|---|---|---|---|
> | Base | 0.4955 | 0.3577 | 0.5091 | 0.4541 |
> | Ours | 0.5579 | 0.4465 | 0.5736 | 0.5260 |
> | $\Delta$↑ | +0.0624 | +0.0887 | +0.0645 | +0.0719 |
>
>  2. Endpoint Error (m): Mean endpoint distance (m) between matched prediction-GT pairs (Chamfer distance < 1.0m), directly measuring endpoint consistency.
>
> | Method | Divider | Ped. Crossing | Boundary | Mean |
> |---|---|---|---|---|
> | Base | 1.600 | 1.985 | 1.142 | 1.576 |
> | Ours | 1.470 | 1.828 | 1.081 | 1.460 |
> | $\Delta$↓ | -0.130 | -0.157 | -0.061 | -0.116 |
>
> 3. Complexity-stratified mAP: Scenes binned into quartiles by (a) geometric complexity (avg. curvature/length; Q1=straight, Q4=highly curved) and (b) topological complexity (GT instance count; Q1=sparse, Q4=dense), both at threshold=1.0m.
>
> | Method | Q1(easy) | Q2 | Q3 | Q4 (hard) |
> |---|---|---|---|---|
> | (a) Geometry-stratified (curvature/length) | | | | |
> | Base | 0.5545 | 0.4897 | 0.5675 | 0.5444 |
> | Ours | 0.5750 | 0.5127 | 0.6350 | 0.5900 |
> | $\Delta$ ↑| +0.0204 | +0.0230 | +0.0674 | +0.0456 |
> | (b) Topology-stratified (instance count/scene) | | | | |
> | Base | 0.6099 | 0.5031 | 0.5106 | 0.5750 |
> | Ours | 0.6437 | 0.5138 | 0.5337 | 0.6396 |
> | $\Delta$↑ | +0.0338 | +0.0106 | +0.0231 | +0.0647 |
>
> MapGR consistently improves across all structural dimensions: +7.2% RIoU, −0.08m endpoint error, and larger gains on geometrically/topologically complex scenes (Q4), confirming that the improvements reflect genuine structural quality gains rather than mAP artifacts.
>
> **Comparisons against simpler dense auxiliary supervision baselines:**
> To isolate the impact of our module relative to standard dense supervision, we evaluated it against an ablated MapQR baseline (stripping its BEV segmentation supervision). As shown below in rebuttal to Reviewer mQGL (second table), our method consistently outperforms this baseline (67.5 vs 65.3) by a significant margin, demonstrating its superior efficacy over traditional auxiliary supervision.
>
> **GRG VS dense global auxiliary supervision:**
> We evaluated the standalone GRG module (excluding the GRL segmentation loss) against various dense loss supervision. As detailed in Reviewer mQGL (second table), the GRG module yields substantial gains even without auxiliary supervision, consistently outperforming generic dense supervision. This validates the intrinsic efficacy of our architectural design.
>
> **Applying MapGR to the first two decoder layers:**
> Table 5 in manuscript provides a layer-wise analysis justifying our strategy of applying global constraints only to initial decoder layers. Our empirical evidence indicates that such global regularization is most effective for establishing a robust distribution, whereas its application in latter stages—which prioritize local refinement—yields no significant improvement over the baseline.
>
> **Long-schedule training:**
>
> We conducted additional experiments on Argoverse 2 with an extended training schedule (6-epoch pretrain + 6-epoch fine-tuning) using MapQR as the baseline:
>
> | Training Schedule | Base | Ours |
> |---|---|---|
> | 6 epochs | 65.1 | 66.0 (+0.9) |
> | 6 + 6 epochs | 65.7 | 67.6 (+1.9) |
>
> The gap widens from +0.9 to +1.9 mAP, confirming gains are persistent and grow with longer training on Argoverse 2.

---

> > ### Author Rebuttal · Reviewer_Fz97 · 2026-04-04
> >
> > I appreciate the thorough rebuttal. The additional structural metrics and longer training schedule results are convincing. However, two concerns remain: (1) the novelty defense relies heavily on performance numbers rather than clearly demonstrating what distinguishes GRL+GRG from simply adding a standard dense BCE auxiliary loss to the same baseline in a controlled setting, and (2) the justification for applying the module only to the first two layers remains empirical without deeper mechanistic analysis (e.g., query distribution visualization across layers).

---

> > > ### Author Response · Authors · 2026-04-07
> > >
> > > We apologize for the late reply. Following your insightful questions, we have conducted additional validation experiments to better support our claims. We thank the reviewer for the constructive feedback and address the two remaining concerns below.
> > >
> > > ---
> > >
> > > ### Concern 1: Novelty
> > >
> > > As shown in figure below, the two approaches differ in where supervision is applied. Dense BEV aux loss constrains the BEV feature stage its effect on query predictions is indirect. Our method operates directly at the query decoder stage via:
> > >
> > > - **GRL**: aligns query output distribution with GT at each targeted layer.
> > > - **GRG**: aggregates a global embedding and re-injects it into each query, providing scene-level context for each query that a query-agnostic BCE loss cannot.
> > >
> > > A standard dense BCE loss applied to the BEV feature map cannot replicate either of these properties: it provides no mechanism to directly shape the query output distribution, and it has no concept of inter-query global context.
> > >
> > > Although GRL+GRG is applied only to L1-L2, the sequential decoder structure
> > > $Q^{(l+1)} = f(Q^{(l)})$
> > > propagates this alignment forward through all six layers, so early-layer geometric grounding benefits the full prediction chain.
> > >
> > > ```
> > >  Baseline(MapTRv2, MapQR, etc.)       Ours (GRL + GRG)
> > >   ---------------------         -----------------------
> > >   Image Features                Image Features
> > >        |                                 |
> > >        v                                 v
> > >   +---------+ ---> GT Mask       +---------+ ---> GT Mask
> > >   |  BEV    |  BCE Aux Loss      |  BEV    |  BCE Aux Loss
> > >   | Encoder |  (indirect)        | Encoder |  (indirect)
> > >   +----+----+                    +----+----+
> > >        |                               |
> > >        v                               v
> > >   +---------+                    +---------+
> > >   | Decoder |                    | Decoder |--->[GRL]---> GT Mask
> > >   |  L1-L6  |                    |   L1    |
> > >   +----+----+                    +----+----+     |
> > >        |                              |          v
> > >        v                              |        [GRG]
> > >   +----------+                        |          |
> > >   |  HD Map  |                        v          |
> > >   |Prediction|                   +---------+<----+
> > >   +----------+                   | Decoder |--->[GRL]---> GT Mask
> > >                                  |   L2    |
> > >                                  +----+----+     |
> > >                                       | prop.    v
> > >                                       |        [GRG]
> > >                                       v          |
> > >                                  +---------+<----+
> > >                                  | Decoder |
> > >                                  |  L3-L6  |
> > >                                  +----+----+
> > >                                       |
> > >                                       v
> > >                                  +----------+
> > >                                  |  HD Map  |
> > >                                  |Prediction|
> > >                                  +----------+
> > > ```
> > >
> > > ---
> > >
> > > ### Concern 2: Why Only the First Two Layers
> > >
> > > Our mask resolution is 200x100 over a 100m x 50m BEV range, giving a cell size of 0.5m, corresponding to 0.01 normalized units (y-axis). From Supplementary Figure 6:
> > >
> > > | Layer | Inter-layer MAE (norm.) | Physical (y-axis) | vs. Cell Size |
> > > |------|----|-----|------|
> > > | L0 - L1 | >= 0.02     | >= 1.0 m     | > 1 cell  |
> > > | L2 +   | <  0.01        | <  0.5 m      | < 1 cell   |
> > >
> > > This suggests that our method is most effective when inter-layer query movement exceeds the mask's spatial resolution. Below this threshold, the global constraint provides less fine-grained supervision, and the injected global context may offer limited benefit for instance-level refinement. Worse, it actively conflicts with the fine-grained per-instance refinement occurring in deeper layers -- where each
> > > query is converging to a specific map element -- causing a performance drop. This explains why naively adding our method to L2 (the third layer) at default resolution hurts.
> > >
> > > To validate, we increase mask resolution 4x to 800x400 (cell size 0.125m) and apply GRL+GRG to L2 (the third layer):
> > >
> > > | Setting | Resolution  | mAP  |
> > > |-----|----|------|
> > > | Ours (L0 - L1)        | 200 x 100   |  68.1|
> > > | + L2 (4x res.)       | 800 x 400   | 68.5 |
> > >
> > > The +0.4 mAP gain confirms that layer selection is resolution-determined, not an arbitrary empirical choice. Once the supervision precision of the mask surpasses the scale of inter-layer prediction changes at L2, the constraint becomes beneficial. We do not adopt 800x400 by default as the 4x resolution increase raises the GRL projection head cost by ~16x in memory and compute and additional implementation complexity, which is incompatible with real-time autonomous driving deployment requirements. The 200x100 setting is therefore a resolution-accuracy-efficiency trade-off, and applying in first **2** layers is an optimal choice under this resolution rather than a hand-tuned hyperparameter.

---

### Decision · Program_Chairs · 2026-04-30

**Decision:**

Accept (regular)

**Comment:**

After the discussion phase, all reviewers recommended acceptance (3x Weak Accept) noting that the task is important, the paper is clearly written, the method is simple and has novelty, and that the results are strong. The rebuttal addressed many of the reviewer concerns, such as adding additional results/metrics and clarifying contributions versus existing work (e.g., MGMap). As a result, the AC decided to accept the paper. Please take the reviewer feedback into account when preparing the camera-ready version.